# Fecal Source Tracking in A Wastewater Treatment and Reclamation System Using Multiple Waterborne Gastroenteritis Viruses

**DOI:** 10.3390/pathogens8040170

**Published:** 2019-09-30

**Authors:** Zheng Ji, Xiaochang C. Wang, Limei Xu, Chongmiao Zhang, Cheng Rong, Andri Taruna Rachmadi, Mohan Amarasiri, Satoshi Okabe, Naoyuki Funamizu, Daisuke Sano

**Affiliations:** 1National Demonstration Center for Experimental Geography Education, School of Geography and Tourism, Shaanxi Normal University, Xi’an 710119, China; jizheng@snnu.edu.cn; 2Key Laboratory of Northwest Water Resource, Ecology and Environment, Ministry of Education, Shaanxi Key Laboratory of Environmental Engineering, Xi’an University of Architecture and Technology, Xi’an 710055, China; xcwang@xauat.edu.cn (X.C.W.); Xulimei@sdau.edu.cn (L.X.); cmzhang@xauat.edu.cn (C.Z.); chenrong@xauat.edu.cn (C.R.); 3Department of Frontier Science for Advanced Environment, Graduate School of Environmental Studies, Tohoku University, Aoba 6-6-06, Aramaki, Aoba-ku, Sendai, Miyagi 980-8579, Japan; andri.rachmadi@kaust.edu.sa; 4Department of Civil and Environmental Engineering, Graduate School of Engineering, Tohoku University, Aoba 6-6-06, Aramaki, Aoba-ku, Sendai, Miyagi 980-8579, Japan; mohan.amarasiri.b5@tohoku.ac.jp; 5Graduate School of Engineering, Hokkaido University, North 13, West 8, Kita-ku, Sapporo, Hokkaido 011-Mizumoto-cho 27-1, Muroran, Hokkaido 060-8628, Japan; sokabe@eng.hokudai.ac.jp; 6Department, Muroran Institute of Technology, Mizumoto-cho 27-1, Muroran, Hokkaido 050-8585, Japan; n_funamizu@mmm.muroran-it.ac.jp

**Keywords:** waterborne gastroenteritis viruses, fecal source tracking, wastewater reclamation, viral contamination

## Abstract

Gastroenteritis viruses in wastewater reclamation systems can pose a major threat to public health. In this study, multiple gastroenteritis viruses were detected from wastewater to estimate the viral contamination sources in a wastewater treatment and reclamation system installed in a suburb of Xi’an city, China. Reverse transcription plus nested or semi-nested PCR, followed by sequencing and phylogenetic analysis, were used for detection and genotyping of noroviruses and rotaviruses. As a result, 91.7% (22/24) of raw sewage samples, 70.8% (17/24) of the wastewater samples treated by anaerobic/anoxic/oxic (A^2^O) process and 62.5% (15/24) of lake water samples were positive for at least one of target gastroenteritis viruses while all samples collected from membrane bioreactor effluent after free chlorine disinfection were negative. Sequence analyses of the PCR products revealed that epidemiologically minor strains of norovirus GI (GI/14) and GII (GII/13) were frequently detected in the system. Considering virus concentration in the disinfected MBR effluent which is used as the source of lake water is below the detection limit, these results indicate that artificial lake may be contaminated from sources other than the wastewater reclamation system, which may include aerosols, and there is a possible norovirus infection risk by exposure through reclaimed water usage and by onshore winds transporting aerosols containing norovirus.

## 1. Introduction

Wastewater treatment and reclamation systems using membrane technologies such as membrane bioreactor (MBR) are becoming increasingly employed in mitigating the shortage of clean water sources [1,2]. However, usage of reclaimed wastewater may increase the exposure risk of humans to pathogenic microorganisms, if the wastewater treatment system is not capable of effectively removing these microorganisms [3]. 

Indicator microorganisms are available to assess and guarantee the microbiological quality of water, because the presence of such indicator microorganisms points to the possible existence of similar pathogens and represents a failure in the treatment system which affects the final effluent [4,5]. Fecal indicator bacteria (FIB) (total coliforms, fecal coliforms, *Escherichia coli*, fecal streptococci and spores of sulphite-reducing *clostridia*) have been used to assess the water quality and treatment performance for decades [5]. However, FIB could not identify the sources of the contamination and there are many complexities related to the extra-enteric ecology of FIBs including environmental persistence and particle association [6,7]. It is unclear how to estimate the contribution of different sources of feces when sources are mixed, which would further hinder the water quality management and health risk evaluation. 

As an alternative, specific microbial source-tracking (MST) markers have been suggested as suitable indicators for evaluating the contamination and treatment performance. crAssphage is one of the suggested human specific contamination markers and found to have geographical and temporal differences [8,9]. *Bacteroidales* and *Lachnospiraceae* which contain host-specific microorganisms are also suggested as alternative indicators [10]. Some studies have suggested waterborne gastroenteritis viruses as MST markers due to their prevalence in host feces and stringent host specificity [11,12,13,14] which provides information on pathogen status that is not provided by indicator bacteria and bacteriophages [6]. 

Even though usage of gastroenteritis viruses as MST markers in evaluating the fecal contamination has been documented, studies in evaluating the suitability of viral indicators to evaluate treatment unit performance are scarce. Especially, in systems like MBR which use size separation as one major virus removal mechanism, microbes with larger diameter sizes (>1 µm), including bacteria (FIB included) and protozoa, can be effectively removed with microfiltration while viral pathogens which are smaller than bacterial pathogens (< 100 nm) could easily pass through the MBR facilities if they are not attached to larger particles, and are much more environmentally resistant than the indicator bacteria [15,16,17,18]. It is further evinced by the absence of correlations between FIB and enteric viruses in MBR effluents [19,20]. Therefore, it is necessary to identify waterborne gastroenteritis viruses circulating in membrane-based wastewater reclamation systems which can be used as indicators to evaluate the treatment unit performance to ensure that reclaimed wastewater is microbiologically safe and not posing infectious risks.

In this study, phylogenetic analysis of multiple waterborne gastroenteritis viruses was applied to estimate contamination sources in a wastewater treatment and reclamation system with a hybrid process of anaerobic/anoxic/oxic (A^2^O) combined with a membrane bioreactor (MBR). Noroviruses and rotaviruses were selected because they were of great significance in disease transmission [21]. The extent of the viral pollution in the system was evaluated by the frequency of positive samples for viral genes from the wastewater samples. The genetic diversity of these viruses was determined by nucleotide sequencing and phylogenetic analysis in order to identify prevalent genotypes and their persistence, which were the underlying evidence for estimating the contamination sources of these gastroenteritis viruses. To the best of our knowledge, a comprehensive study of this kind, by the inclusion of human viruses in wastewater, has rarely before been performed in northwestern China.

## 2. Results

### 2.1. Occurrence of Viral Genes in Wastewater Samples 

We analyzed the quantity of human norovirus GI, GII and rotavirus and their removal in a wastewater treatment plant utilized in a University Campus. Wastewater influent contained septic tank effluents, kitchen wastewater and greywater. Wastewater was treated using fine screen, A^2^O treatment and MBR. Effluent wastewater was discharged in to a recreational lake. 

Concentration of complex environmental samples might also simultaneously concentrate the PCR inhibitory substances, thus resulting in interference in virus detection. To increase sensitivity, the nested/semi-nested PCR was employed. The results of inhibition test indicated that PCR inhibitors possibly existing in wastewater did not affect the virus detection from the collected samples (data not shown). The occurrences of viruses in samples collected from different sites were summarized in Table 1. High level of fecal contamination in the study area was revealed by the high percentages of positive samples for norovirus and rotavirus. After analyzing 96 wastewater samples, norovirus GI and GII were found in 52% (50/96) and rotavirus in 32% (31/96) of samples (Table 1).

The number of viruses detected in wastewater samples from different sites was variable. Only one virus was detected in 16% (15/96) of samples, including 5 raw sewage samples, 4 A^2^O effluent samples and 6 lake water samples. More than one virus type was found in 29% (28/96) of samples, including 16 raw sewage samples, 7 A^2^O effluent samples, and 5 lake water samples. These indicate that different families of gastroenteritis viruses are co-circulating in the study area. For mixed raw sewage collected after the fine screen, 22 samples (92%) were positive for viruses; norovirus GI/GII was found in 83% (20/24) and rotavirus in 75% (18/24). Gastroenteritis viruses in raw sewage must have originated from black water from toilet flushing and grey water from washing, which are potentially contaminated by feces or vomit from infected humans. For the A^2^O effluent samples, 17 samples representing 71% (17/24) were positive; norovirus were found in 71% (17/24) and rotavirus in 29% (7/24). For lake water, 14 (58%) samples were positive for viruses. Norovirus was found in 54.2% (13/24) while rotavirus was found in 25.0% (6/24).

### 2.2. Phylogenetic Analysis of Norovirus 

The norovirus sequences detected in wastewater samples were distributed between the two genogroups. 72% (36/50) of the sequences were similar to GI while 78% (39/50) belonged to GII, whereas 50% (25/50) of them were positive for both GI and GII. Figure 1 and Figure 2 illustrate the result of phylogenetic analysis for capsid region in norovirus genes obtained from wastewater samples. Multiple genotypes of norovirus (GI.3, GI.4, GI.6, GII.3, GII.4 (Den Haag), GII.6 and GII.13) circulating in the study area between human populations and wastewater were detected. The high similarity in identities between norovirus genes detected from multiple samples collected from different sampling sites in this area might suggest that the samples might be contaminated by human noroviruses from the same original source—the residents in the study area.

### 2.3. Molecular Detection and Characterization of Rotavirus

Group A rotavirus has been shown to be the most prevalent rotavirus in children and adults over the world [22,23]. Therefore, these viruses are considered of great epidemiological importance. Human rotaviruses (HRVs) were characterized with genotype-specific primers for VP7 (G genotype). The phylogenetic analysis was performed for the PCR products derived from wastewater samples (Figure 3), which indicated that all clones were highly homologous to human rotavirus isolates. The most frequent G type detected was type G9, followed by G2 and G3. 

## 3. Discussion

We confirmed the presence of human noroviruses (GI and GII) and rotaviruses in the influent wastewater, fine screen effluent, A^2^O treatment effluent, and the lake water receiving the wastewater effluents. The lower virus detection rate observed after the A^2^O treatment process compared to raw sewage may be owing to the attachment to wastewater solids and the presence of antiviral components in the activated sludge [24,25,26,27]. Gastroenteritis viruses were not detectable in the samples of MBR effluent after free chlorine disinfection. MBR combined with chlorine treatment may have significantly contributed to the reduction of virus particles, or at least the MBR with chlorine treatment may decrease the virus quantity to a very low extent which was below the detection limit [28]. However, 54% of the lake water samples were positive for viruses, implying that the MBR effluent disinfected with free chlorine may not be the source of virus contamination in the lake water. 

The results of phylogenetic analysis revealed that the artificial lake was contaminated by multiple human viruses. In this case, sewage pipe leakage and overflows are not likely to cause such contamination due to the adequately designed capacity and the proper maintenance of the water distribution system. Secondary contamination of lake water may occur from unidentified nonpoint sources. As the lakes are open water bodies in the local water system, they were vulnerable to contamination generating from natural processes (such as surface runoff, water air transfer and wild animals) or human activities [29,30]. As non-point sources of gastroenteritis viruses, rain water inflow and aerosol blowing into the lakes may be considered as possible reasons. Furthermore, it would be of particular concern because the microbial aerosols containing viral particles could be formed during water reclamation, and exposure to reclaimed water can pose a potential health risk [31]. On the other hand, onshore winds around 4 m/s can contain 5.3 ± 1.2 × 10^4^ m^−3^ of viruses [32]. These results underscore the possible impact of viral exposure by reclaimed water consumption, and by being exposed to winds containing aerosols and suggests that the control of non-point viral sources, and storage and safe use of reclaimed water should be the focus of wide attention. 

The sequence diversity of human noroviruses, especially for the capsid region, from environmental samples has been reported in several studies [33,34,35]. The isolation of both GI and GII strains in this study would indicate the co-existence of extensive recessive infections for both genogroups which may not be included and documented in previous epidemiological surveys. However, results similar to our present study have been obtained in some environmental studies [36,37]. Thus it might indicate a distinct genogroup prevalent bias between clinical samples and environmental samples [38,39]. It has been demonstrated that the viral loads of GI in fecal samples was reported less than one percent of that of GII and GI is generally more resistant to wastewater treatment and disinfection than GII [38,39], suggesting the differences in environmental occurrence and persistence of GI and GII strains [40]. Although there was no documentation about the viral infection in the studying area, the report of Xi’an Center for Disease Control and Prevention showed that HuNoV GII was more prevalent than HuNoV GI in clinical samples (data not shown). However, human norovirus strains detected in wastewater may reflect more accurate actual circulation among population rather than clinical survey, because wastewater receive viruses shed from patients with both symptomatic and asymptomatic infections. Thus, the findings indicate the possibility that norovirus GI strains might be more widely spread among humans than previously thought. Other explanations such as seasonal or geographic variation in viral RNA levels could not be excluded either. 

Number of rotavirus A genotypes (G1, G2, G3, and G9) were detected during the sampling period and G9 was predominant. Previous surveys confirmed the circulation of multiple rotavirus A genotypes in the same area in the same year [22] even though the predominant rotavirus genotype varied in different geographical regions [41,42,43]. The phylogenetic analysis of rotavirus also suggests that the viruses detected in this study might originate from infant, children or healthy carriers, and thus their contamination sources or transport routes could be different from those of fecal indicators usually originating from adults.

It has been recognized that enteric viruses are more stable than indicator bacteria in water and sewage, constituting not only a potential hazard but also a good tracer for fecal pollution source tracking [14,44,45]. Wastewater treatment plants (WWTPs) have played an important role in microbiological reduction, minimizing the risks associated with pathogen circulation into the environment [3,18]. However, little is known about the comparative persistence or survival of source-specific markers and strains, and the available data for markers ranging from *E. coli* to *Bacteroidales* and phage markers indicate strongly that survival is not proportional [46]. The general trend is that the dominance of environmental strains differ from strains in the host. Due to the inherent difficulty in finding a correlation between environmental contamination and cases of infection, microbiological monitoring of the environment might be more helpful for source tracking and water safety control rather than risk assessment [47,48]. In addition, limited waterborne viral outbreaks usually occurred at distance from the original source of contamination. This study provides novel evidence of the prevalence and genetic diversity of waterborne gastroenteritis viruses and the potential of human noroviruses for microbial source tracking due to its host-specificity and higher sensitivity of (semi-)nested PCR (detection about 10^0^ copies/reaction) [49,50]. Attention should be paid to the emerging health threat due to the different predominant types of the targeting viruses observed in the study. 

Furthermore, although direct sequencing analysis with well-purified PCR amplicons could be useful for providing information on viral identification in wastewater [37], the potential that the results may have a bias in interpreting the genetic diversity of the viral types might not be neglected. This might be resulted from the inhibition effect as the recovery rate of water concentration [3] and the affinity selection of PCR reaction might be type and strain different for viruses [51]. This more comprehensive analysis of the relative abundance and occurrence of viruses in wastewaters may allow for the development of more conservative viral tracers and complementary indicators to further ensure the microbial safety of wastewater reclamation systems.

## 4. Materials and Methods 

### 4.1. Sample Collection

To investigate waterborne gastroenteritis viral pollution, four kinds of wastewater samples were collected four times per month for a 6-month sampling period (from Feb. to Jul., 2012, the total sample number is 96) in a wastewater treatment and reclamation system in Xi’an Si-yuan University. The university is located in the south-eastern suburb of Xi’an in Northwest China. WWTP is a hybrid of anaerobic/anoxic/oxic (A^2^O) combined with a membrane bioreactor (MBR) (As shown in Figure 4) [52,53]. The influent is a mixture of black water from toilet flushing, grey water from miscellaneous uses, and kitchen wastewater from the university canteens. The reclaimed water is supplied to the lakes in the campus which have both the functions of landscaping and storage reservoirs where the water is further supplied to buildings for toilet flushing and/or to the green belt for gardening and irrigation. All samples were collected on clear weather days, stored in sterilized plastic bottles on ice, and delivered to the laboratory within several hours after collection.

### 4.2. Recovery of Viral Particles and Nucleic Acid Extraction 

Since the density of waterborne gastroenteritis viruses is presumed to be very low in water, an efficient viral concentration method is required [49]. It is important to recognize that there is no single method yet by which it is possible to recover all enteric viruses with high efficiency from diverse types of water samples [49]. On the basis of the properties of urban sewage and viral particles, the methods of aqueous polymer two-phase separation (polyethylene glycol precipitation, PEG precipitation) and/or virus adsorption elution (VIRADEL) using electronegative membrane filters (mixed cellulose ester) were applied to concentrate viruses from different types of wastewater samples in the study [49]. For high turbidity (>100 NTU) samples such as raw sewage collected after the fine screen and the effluent of A^2^O treatment tank, 250 mL of each was concentrated by PEG precipitation method [54,55]. For low turbidity (<100 NTU) samples such as the effluent of MBR and the lake water, 2 L of each was concentrated by VIRADEL method [56] followed by PEG precipitation. Viral concentrates were resuspended in 1 mL distilled deionized water (DDW) and immediately processed for nucleic acid extraction or stored at −80 °C until use.

Viral RNA was extracted from sample concentrates with QIAamp^®^ Viral RNA Mini Kit (Qiagen, Hilden, Germany), following the manufacturer’s instructions. Complementary DNA (cDNA) was synthesized from 10 μL out of 60 μL of the extracted RNA with DNase treatment and subsequent reverse transcription (RT) reaction using PrimeScript^®^ RT reagent Kit with gDNA Eraser (Takara, Dalian, China) according to the protocol described by the manufacturer. The synthesized cDNA was stored at −80 °C for further analysis.

### 4.3. Molecular Detection and Characterization of Enteric Viruses

The detection and characterization of waterborne gastroenteritis viruses were performed with a combination of several molecular techniques which allowed both sensitive and precise identification of predominant human pathogenic viruses occurring in urban sewers. The capsid encoding region with higher host-specificity was chosen for nested or semi-nested PCR detection of HuNoVs and HRVs (Table 2). The molecular characterization of HuNoVs and HRVs was performed by sequencing and phylogenetic analysis of the second round of PCR amplicons. For the first PCR round, 2 μL of cDNA was added to a reaction mixture consisting of 0.25 μL of Ex Taq (Takara, Dalian, China), 2.5 μL of 10× Ex Taq Buffer, 2 μL of deoxynucleoside triphosphate (dNTP) mixture, and 400 nM of each PCR primer, and all mixed with DDW to obtain a total volume of 25 μL. For the second PCR round, the same concentration of reagents was used with 2 μL of 1000-fold dilution of the first PCR product added to the PCR tube. Primer sequences and positions, and cycling conditions for detection and characterization of each viral group are shown in Table 1. Positive and negative controls (clinical samples for each virus type and RNA/DNA-free water) were included in all PCR runs. PCR products were analyzed by gel electrophoresis on a 1.5% (wt/vol) strength agarose gel, stained with GelRed^TM^ Nucleic Acid gel stain (Biotium, Fremont, CA, USA), and visualized by UV illumination. When no amplification products were observed, two-fold and four-fold dilutions of the identical wastewater sample were prepared and applied to the nested/semi-nested RT-PCR for checking the presence of PCR inhibition. As the reference, 1 mL of DDW added with 1 μL virus suspension was used.

### 4.4. Nucleotide Sequencing and Phylogenetic Analysis

PCR products obtained from the second round of amplification for each virus group were excised from the gel and purified immediately. The purified nucleotides were sent to Sangon Biotech (Shanghai, China) Co., Ltd for sequence determination. After checking the sequence chromatograms with Chromas software (version 2.31) for errors, the final sequences were obtained. Homology searches were conducted using the GenBank server of the National Centre for Biotechnology Information (NCBI) and the Basic Local Alignment Search Tool (BLAST) algorithm and calicivirus typing tool (https://norovirus.phiresearchlab.org/). Phylogenetic relationships were generated using maximum likelihood method using MEGA 7 by Kimura 2-parameter model with nucleotide substitution rates following a gamma-distribution. One thousand bootstrap replications were performed to evaluate the robustness of each node [60,61,62]. Interactive Tree Of Life (iTOL) v4 was used to develop the phylogenetic trees [63]. 

### 4.5. Nucleotide Sequence Accession Numbers 

The nucleotide sequences corresponding to fragments of rotaviruses and noroviruses have been deposited in the GenBank database under accession No. KF854668 to KF854698 and KF854593 to KF854667, respectively. 

## 5. Conclusions

In conclusion, this study describes novel findings on the prevalence and genetic diversity of human gastroenteritis viruses in water in China. It confirmed that human fecal contamination is widespread and also that viral tools are applicable as fecal indicators and tracers in all geographical areas studied. Continuous viral contamination monitoring is useful for preventing waterborne disease outbreaks and for understanding the impact caused by human activities and the use of reclaimed wastewater.

Furthermore, this study highlights the importance of further environmental studies toward a better understanding of the circulation of gastroenteritis viruses in aquatic environments and human populations. In other words, circulation of gastroenteritis viruses between contaminated environmental water and human populations is a key issue in understanding their epidemiology and health risks for humans. Further studies are needed to define the relationship between the level of gastroenteritis viruses contamination detected by PCR in reclaimed wastewater and the potential effect and health risk of these wastewater after consumption.

## Figures and Tables

**Figure 1 pathogens-08-00170-f001:**
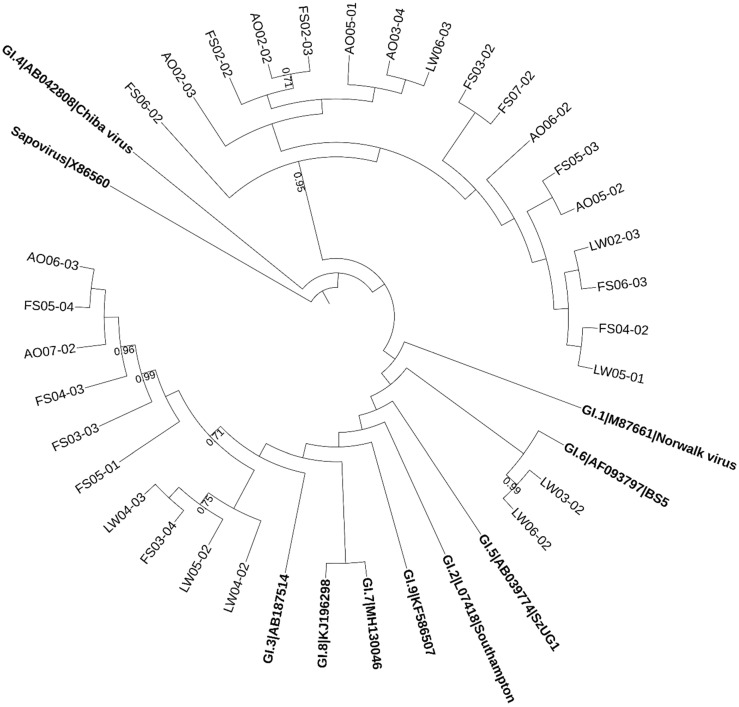
The phylogenetic tree based on partial sequences of the capsid gene of norovirus GI. The tree was constructed by the maximum-likelihood method with 1000 bootstrap replicates using MEGA7 and depicted using iTOL4. The obtained sequences were expressed as the abbreviation of sampling site + month + time. Numbers at each branch indicate bootstrap values for the clusters supported by that branch (>0.7). Numbers at each branch indicate bootstrap values for the clusters supported by that branch. Sapovirus was used as an out group. Reference sequences are shown in bold face.

**Figure 2 pathogens-08-00170-f002:**
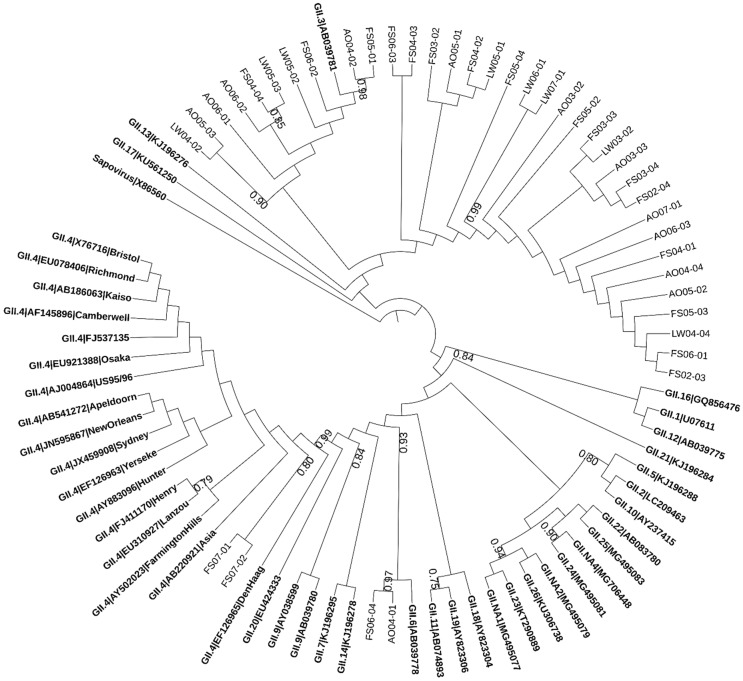
The phylogenetic tree based on partial sequences of the capsid gene of norovirus GII. The tree was constructed by the maximum-likelihood method with 1000 bootstrap replicates using MEGA7 and depicted using iTOL4. The obtained sequences were expressed as the abbreviation of sampling site + month + time. Numbers at each branch indicate bootstrap values for the clusters supported by that branch (>0.7). Numbers at each branch indicate bootstrap values for the clusters supported by that branch. Sapovirus was used as an out group. Reference sequences are shown in bold face.

**Figure 3 pathogens-08-00170-f003:**
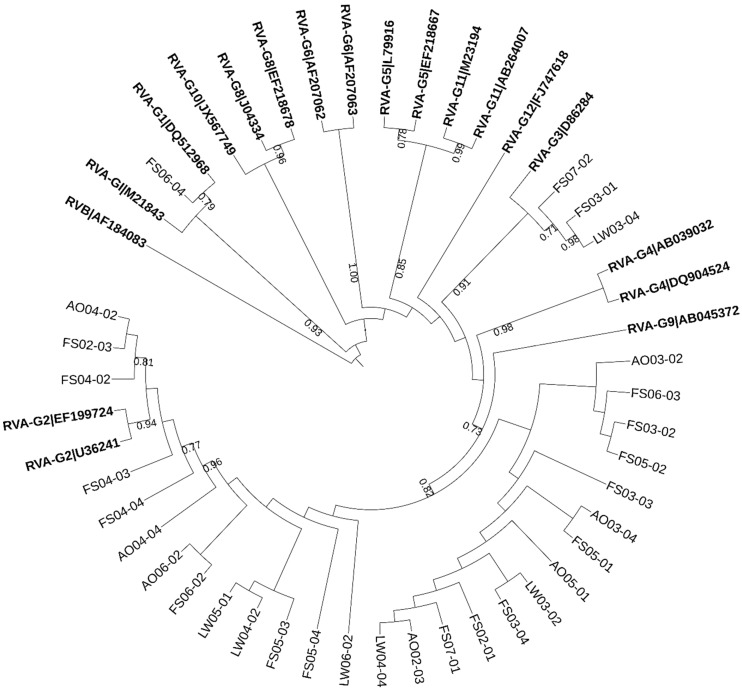
The phylogenetic tree based on partial sequences of the VP7 gene of rotavirus. The tree was constructed by the maximum-likelihood method with 1000 bootstrap replicates using MEGA7 and depicted using iTOL4. The obtained sequences were expressed as the abbreviation of sampling site + month + time. Numbers at each branch indicate bootstrap values for the clusters supported by that branch (>0.7). Numbers at each branch indicate bootstrap values for the clusters supported by that branch. Human Rotavirus B (RVB) was used as an out group. Reference sequences are shown in bold face.

**Figure 4 pathogens-08-00170-f004:**
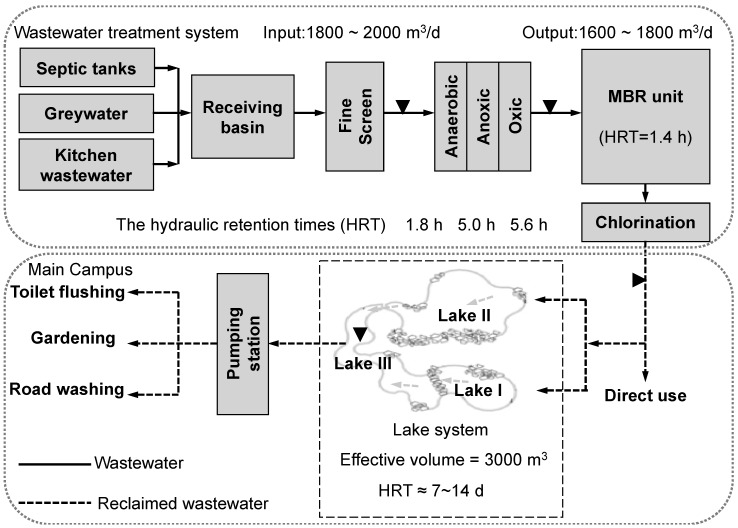
Sampling locations in the local wastewater treatment and reclamation system. Four types of wastewater samples were mixed raw sewage samples collected after fine screen (FS), the effluent of A^2^O treatment tank (AO), MBR effluent after disinfection (MBR) and lake water (LW).

**Table 1 pathogens-08-00170-t001:** Occurrences of waterborne gastroenteritis viruses in wastewater samples.

Virus	Sampling Locations % (Positive/Total Samples)				Total Detection Rate for Each Virus (%)
	Mixed Raw Sewage	A^2^O Effluent	MBR Effluent after Disinfection	Lake Water	
HuNoV GI	67 (16/24)	45 (11/24)	0 (0/24)	38 (9/24)	38 (36/96)
HuNoV GII	79 (19/24)	50 (12/24)	0 (0/24)	33 (8/24)	41 (39/96)
HRVs	75 (18/24)	29 (7/24)	0 (0/24)	25 (6/24)	32 (31/96)
Total Detection Rate for Each Sampling Site (%)	92 (22/24)	71 (17/24)	0 (0/24)	63 (15/24)	56 (54/96)

**Table 2 pathogens-08-00170-t002:** Primers and amplification conditions used for detection and molecular characterization of waterborne gastroenteritis viruses.

Virus	Target Gene	PCR Round	Primer	Sequence (5’-3’) ^a^	Reference
Rotavirus	VP7(G)	1st	RoA ^b^	CTTTAAAAGAGAGAATTTCCGTCTG	[57,58]
		1st	RoB ^b^	TGATGATCCCATTGATATCC	
		2nd	RoC ^b^	TGTATGGTATTGAATATACCAC	
		2nd	RoD ^b^	ACTGATCCTGTTGGCCAWCC	
Norovirus GI	ORF1–ORF2 junction	1st	COG1F ^c^	CGYTGGATGCGNTTYCATGA	[34,59]
		1st	G1-SKR ^c^	CCAACCCARCCATTRTACA	
		2nd	G1-SKF ^c^	CTGCCCGAATTYGTAAATGA	
		2nd	G1-SKR ^c^	CCAACCCARCCATTRTACA	
Norovirus GII	ORF1–ORF2 junction	1st	COG2F ^d^	CARGARBCNATGTTYAGRTGGATGAG	[34,59]
		1st	G2-SKR ^e^	CCRCCNGCATRHCCRTTRTACAT	
		2nd	G2-SKF ^e^	CNTGGGAGGGCGATCGCAA	
		2nd	G2-SKR ^e^	CCRCCNGCATRHCCRTTRTACAT	

^a^ Mixed bases in degenerate primers are as follows: K = G/T; M = A/C; R = A/G; S = G/C; W = A/T; Y = C/T; B = G/T/C; H = A/T/C; N = A/T/G/C; ^b^ Corresponding nucleotide position of HRV (K02033) of the 5’ end; ^c^ Corresponding nucleotide position of HuNoV (M87661) of the 5’ end; ^d^ Corresponding nucleotide position of HuNoV (AF145896) of the 5’ end; ^e^ Corresponding nucleotide position of HuNoV (X86557) of the 5’ end. Rotavirus, 1^st^ PCR: 94 °C for 3 min; 35 cycles of 94 °C for 30 s, 37 °C for 30 s, and 72 °C for 1 min; and 72 °C for 5 min; 2^nd^ PCR: 94 °C for 3 min; 35 cycles of 94 °C for 30 s, 37 °C for 30 s, and 72 °C for 30 s; and 72 °C for 5 min. Norovirus, 94 °C for 5 min; 40 cycles of 94 °C for 30 s, 50 °C for 30 s, and 72 °C for 30 s; and 72 °C for 10 min.

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
