# Peer review of "Fecal Source Tracking in A Wastewater Treatment and Reclamation System Using Multiple Waterborne Gastroenteritis Viruses"

_pathogens, 2019, doi:10.3390/pathogens8040170_

Round 1

Reviewer 1 Report

In this article, the authors correctly identified a major gap in the identification of specific pathogens in contaminated wastewater i.e. that identifying fecal indicator bacteria do not identify the sources of contamination nor how to estimate the contribution of different sources of feces when sources are mixed. The have applied an alternative method of microbial source tracking (MST) markers to specifically identify noroviruses and rotaviruses using phylogenetic analyses to determine the genetic diversity of these viruses.

Here are some issues that the authors must address:

Line 33: "...samples collected from MBR..."; please define 'MBR' here since that's the first occurrence in the manuscript

Line 45: "...exposure risk of human..."; change to "humans"

Line 48: "...existence of a similar pathogens..."; delete "a"

Line 64: "(...which uses..."; change to "which use.."

Line 75: change "combining" to "combined"

Lines 85- 86: Please re-write sentence. Confusing particularly the part following "as well". Did authors intend to say that "concentration of complex environmental samples might also concentrate the PCR inhibitory substances, thus resulting in interference in virus detection?"

Line 87: "The results of the inhibition test..."; Describe the inhibition test, what steps were taken to address inhibition?

Line 90: "...of positive samples, it presented..."; define "it". I'm assuming that "it" referes to "the occurrences of viruses recovered from the collection sites" in previous sentence?

Line 92: "...norovirus GI and GI"; should that be "GI and GII?" Please correct

Line 106: "...norovirus were..."; change to "norovirus was"

Line 139: Define 'HRVs"; First instance of the abbreviation

Line 184: "...GI infection were..."; change to  "GI infection was"

Line 185: "...human noroviruses strains..."; change to "human norovirus strains"

Lines 198-200: Need a reference for this statement

Lines 205-208: Excellent point

Lines 211-212: "The attention..."; Delete 'The"

Line 222: delete "in order" and "the"

Line 225: change "combining" to "combined"

Line 227: "...from the university canteens"; Which university? Where is  this university located? Proximity?

Lines 262-263: Define HAstVs and HuSaVs

Author Response

We thank you very much for giving us an opportunity to revise our manuscript, and appreciate the  reviewers for their positive and constructive comments and suggestions on our manuscript. We have clarified the points raised by the reviewers as follows.

Reviewer

In this article, the authors correctly identified a major gap in the identification of specific pathogens in contaminated wastewater i.e. that identifying fecal indicator bacteria do not identify the sources of contamination nor how to estimate the contribution of different sources of feces when sources are mixed. The have applied an alternative method of microbial source tracking (MST) markers to specifically identify noroviruses and rotaviruses using phylogenetic analyses to determine the genetic diversity of these viruses.

Here are some issues that the authors must address:

Comment: Line 33: "...samples collected from MBR..."; please define 'MBR' here since that's the first occurrence in the manuscript

Response: Modified as suggested (Line 45)

Comment: Line 45: "...exposure risk of human..."; change to "humans"

Response: Modified as suggested (Line 47)

Comment: Line 48: "...existence of a similar pathogens..."; delete "a"

Response: Modified as suggested (Line 51)

Comment: Line 64: "(...which uses..."; change to "which use.."

Response: Modified as suggested (Line 69)

Comment: Line 75: change "combining" to "combined"

Response: Modified as suggested (Line 81)

Comment: Lines 85- 86: Please re-write sentence. Confusing particularly the part following "as well". Did authors intend to say that "concentration of complex environmental samples might also concentrate the PCR inhibitory substances, thus resulting in interference in virus detection?"

Response: We modified as suggested (Line 95-96)

Comment: Line 87: "The results of the inhibition test..."; Describe the inhibition test, what steps were taken to address inhibition?

Response: It is described in line 289-291.

As we focused on genotyping of gastroenteritis viruses, the dilutions of negative samples of (semi-)nested PCR were used for inhibition test only in order to check the inhibition effect. It means that only qualitative PCR were applied in this experiment in order to know that if dilution could address the PCR inhibition.

Comment: Line 90: "...of positive samples, it presented..."; define "it". I'm assuming that "it" referes to "the occurrences of viruses recovered from the collection sites" in previous sentence?

Response: Paragraph was modified for clarification (Line 99-102)

Comment: Line 92: "...norovirus GI and GI"; should that be "GI and GII?" Please correct

Response: Modified as suggested (Line 102)

Comment: Line 106: "...norovirus were..."; change to "norovirus was"

Response: Modified as suggested (Line 114)

Comment: Line 139: Define 'HRVs"; First instance of the abbreviation

Response: Modified as suggested (Line 147)

Comment: Line 184: "...GI infection were..."; change to  "GI infection was"

Response: Section was modified (Line 191-202).

Comment: Line 185: "...human noroviruses strains..."; change to "human norovirus strains"

Response: Section was modified (Line 191-202).

Comment: Lines 198-200: Need a reference for this statement

Response: We added the references (Line 212).

Comment: Lines 205-208: Excellent point

Response: Thank you very much for highlighting.

Comment: Lines 211-212: "The attention..."; Delete 'The"

Response: Modified as suggested (Line 224)

Comment: Line 222: delete "in order" and "the"

Response: Modified as suggested (Line 237)

Comment: Line 225: change "combining" to "combined"

Response: Modified as suggested (Line 241)

Comment: Line 227: "...from the university canteens"; Which university? Where is this university located? Proximity?

Response: Description was added in line 239-242.

Comment: Lines 262-263: Define HAstVs and HuSaVs

Response: We deleted them in this version.

Reviewer 2 Report

Overall feedback:  This study is an important addition to the science surrounding the management of water quality and the control of waterbourne disease.  The viral approach is unique and very much needed, as most discussions of water pathogens are limited to bacteria.  

I would suggest deepening your references to better highlight the importance of finding alternative sewage indicators, including:

O’Mullan GD, Dueker ME, Juhl AR. 2017. Challenges to Managing Microbial Fecal Pollution in Coastal Environments: Extra-Enteric Ecology and Microbial Exchange Among Water, Sediment, and Air. Current Pollution Reports.3:1-16.

McLellan SL, Eren AM. 2014. Discovering new indicators of fecal pollution. Trends Microbiol.22:697-706.

VandeWalle JL, Goetz GW, Huse SM, Morrison HG, Sogin ML, Hoffmann RG, Yan K, McLellan SL. 2012. Acinetobacter, Aeromonas and Trichococcus populations dominate the microbial community within urban sewer infrastructure. Environ Microbiol. Sep;14:2538-2552.

Also, I am intrigued by your assertion that the contamination of the lake water is due to external influences including potential aerosolization -- I think you need to make a stronger case for this (you need to provide a bit more information regarding why you are convinced that it must be external, you should provide more references illustrating the possibility, etc.).  You may want to consider the following references as you do so:

This one measures viral aerosols:

Dueker ME, O’Mullan GD, Martinez J, Juhl AR, Weathers KC. 2017. Onshore wind speed modulates microbial aerosols along an urban waterfront. Atmosphere. 2017-11-09;8:215.

This one looks at the exchange of microbes between water and air:

Dueker ME, French S, O’Mullan GD. 2018. Comparison of Bacterial Diversity in Air and Water of a Major Urban Center. Frontiers in Microbiology.9.

Finally, it is very hard to follow the results and discussion without a clear idea of the layout of the facility -- is it possible to include the site map before the methods?  Or at least a better description of the lakes when referencing them would be helpful.  

Line by line feedback:

--the title is unclear -- maybe change to "Fecal Source Tracking in a Wastewater Treatment and Reclamation System Using Multiple Waterborne Gastroenteritis Viruses"

--Line 33 -- define MBR

--Lines 35-37 are not obviously supported by abstract -- need to be better contextualized in the abstract itself

--Line 45 -- should be "humans"

--line 92 -- remove right parenthesis

--Table 1 - are these all percentages?  Need more explanation of what the table is communicating

--Line 116 -- do you mean "high similarity in identities"?

--Line 156 -- what do you mean by "strong bacterial activities"?

--Line 181-188 -- can you provide data to clearly show how GI and GII differ in your study?  It would make this easier to follow and more convincing.

--Line 188 -- I am not sure you can say this based on your study -- please provide more context and connection to your study's findings 

--Line 211 -- you need to provide stronger evidence of "higher sensitivity"

--Lines 214-219 -- not sure they are necessary -- please provide more context if you keep these -- for instance, the first sentence is hard to follow -- do you mean that there is bias?  or that it is neglected? 

Author Response

We thank you very much for giving us an opportunity to revise our manuscript, and appreciate the  reviewers for their positive and constructive comments and suggestions on our manuscript. We have clarified the points raised by the reviewers as follows.

Reviewer

Overall feedback:  This study is an important addition to the science surrounding the management of water quality and the control of waterborne disease.  The viral approach is unique and very much needed, as most discussions of water pathogens are limited to bacteria. 

Comment: I would suggest deepening your references to better highlight the importance of finding alternative sewage indicators, including:

O’Mullan GD, Dueker ME, Juhl AR. 2017. Challenges to Managing Microbial Fecal Pollution in Coastal Environments: Extra-Enteric Ecology and Microbial Exchange Among Water, Sediment, and Air. Current Pollution Reports.3:1-16.

McLellan SL, Eren AM. 2014. Discovering new indicators of fecal pollution. Trends Microbiol.22:697-706.

VandeWalle JL, Goetz GW, Huse SM, Morrison HG, Sogin ML, Hoffmann RG, Yan K, McLellan SL. 2012. Acinetobacter, Aeromonas and Trichococcus populations dominate the microbial community within urban sewer infrastructure. Environ Microbiol. Sep;14:2538-2552.

Response: Thank you very much for the suggestion. We have modified the introduction as suggested (Line 54-56, 62-63).

Comment: Also, I am intrigued by your assertion that the contamination of the lake water is due to external influences including potential aerosolization -- I think you need to make a stronger case for this (you need to provide a bit more information regarding why you are convinced that it must be external, you should provide more references illustrating the possibility, etc.).  You may want to consider the following references as you do so:

This one measures viral aerosols:

Dueker ME, O’Mullan GD, Martinez J, Juhl AR, Weathers KC. 2017. Onshore wind speed modulates microbial aerosols along an urban waterfront. Atmosphere. 2017-11-09;8:215.

This one looks at the exchange of microbes between water and air:

Dueker ME, French S, O’Mullan GD. 2018. Comparison of Bacterial Diversity in Air and Water of a Major Urban Center. Frontiers in Microbiology.9.

Response: We modified our discussion with additional references (Line 171-184).

Comment: Finally, it is very hard to follow the results and discussion without a clear idea of the layout of the facility -- is it possible to include the site map before the methods?  Or at least a better description of the lakes when referencing them would be helpful. 

Response: We added a description at the beginning of the results section (Line 91-94).

Line by line feedback:

Comment: --the title is unclear -- maybe change to "Fecal Source Tracking in a Wastewater Treatment and Reclamation System Using Multiple Waterborne Gastroenteritis Viruses"

Response: Modified as suggested

Comment: --Line 33 -- define MBR

Response: Modified as suggested (Line 45)

Comment: --Lines 35-37 are not obviously supported by abstract -- need to be better contextualized in the abstract itself

Response: We modified the abstract to clarify the idea (Line 34 – 39)

Comment: --Line 45 -- should be "humans"

Response: Modified as suggested (Line 47)

Comment: --line 92 -- remove right parenthesis

Response: Modified as suggested

Comment: --Table 1 - are these all percentages?  Need more explanation of what the table is communicating

Response: Table 1 was modified.

Comment: --Line 116 -- do you mean "high similarity in identities"?

Response: Corrected as suggested (Line 123-126).

Comment: --Line 156 -- what do you mean by "strong bacterial activities"?

Response: Clarified the sentence (Line 163-165).

Comment: --Line 181-188 -- can you provide data to clearly show how GI and GII differ in your study?  It would make this easier to follow and more convincing.

Response: We modify the Table 1 to separately report GI and GII detection data.

Comment: --Line 188 -- I am not sure you can say this based on your study -- please provide more context and connection to your study's findings

Response: We modified the expression (Line 196-202).

Comment: --Line 211 -- you need to provide stronger evidence of "higher sensitivity"

Response: New references were added and the description was modified (Line 221-226).

Comment: --Lines 214-219 -- not sure they are necessary -- please provide more context if you keep these -- for instance, the first sentence is hard to follow -- do you mean that there is bias?  or that it is neglected?

Response: Modified as suggested (Line 227-234).

Round 2

Reviewer 2 Report

This paper is much improved.  I do think it needs a bit more editing in terms of language, but generally fine.  Just one suggestion:  the insertion of "epidemiological study" in the abstract does not make sense to me -- I associate epidemiology with human health, not science conducted using molecular tools......?

Author Response

Comment: This paper is much improved.  I do think it needs a bit more editing in terms of language, but generally fine.  Just one suggestion:  the insertion of "epidemiological study" in the abstract does not make sense to me -- I associate epidemiology with human health, not science conducted using molecular tools......?

Response: The term "epidemiological study" was removed according to the reviewer's suggestion.